# Prospective Comparison of Handheld Ultrasound Devices from Different Manufacturers with Respect to B-Scan Quality and Clinical Significance for Various Abdominal Sonography Questions

**DOI:** 10.3390/diagnostics13243622

**Published:** 2023-12-08

**Authors:** Daniel Merkel, Tim Felix Züllich, Christoph Schneider, Masuod Yousefzada, Diana Beer, Michael Ludwig, Andreas Weimer, Julian Künzel, Roman Kloeckner, Johannes Matthias Weimer

**Affiliations:** 1BIKUS—Brandenburg Institute for Clinical Ultrasound, Brandenburg Medical School Theodor Fontane (MHB), 16816 Neuruppin, Germany; daniel.merkel@mhb-fontane.de; 2Immanuel Klinik Rüdersdorf, University Hospital of the Brandenburg Medical School, 15562 Rüdersdorf bei Berlin, Germany; tim.zuellich@immanuelalbertinen.de (T.F.Z.); christoph.schneider@immanuelalbertinen.de (C.S.); masuod.yousefzada@immanuelalbertinen.de (M.Y.); diana.beer@immanuelalbertinen.de (D.B.); 3Department of Internal Medicine I, Hospital of the German Armed Forces Berlin, 10115 Berlin, Germany; michael6ludwig@bundeswehr.org; 4Center of Orthopedics, Trauma Surgery and Spinal Cord Injury, Heidelberg University Hospital Heidelberg, 69118 Heidelberg, Germany; andreas.weimer@kkh-bergstrasse.de; 5Department of Otorhinolaryngology, Head and Neck Surgery, University Hospital Regensburg, 93053 Regensburg, Germany; julian.kuenzel@klinik.uni-regensburg.de; 6Institute of Interventional Radiology, University Hospital Schleswig-Holstein—Campus Lübeck, 23538 Lübeck, Germany; roman.kloeckner@uksh.de; 7Rudolf Frey Learning Clinic, University Medical Center of the Johannes Gutenberg University Mainz, 55131 Mainz, Germany

**Keywords:** B-scan, sonography, quality, comparison image quality, HEUS, high-end ultrasound, HHUS, handheld ultrasound, pocket ultrasound

## Abstract

Background: Handheld ultrasound (HHUS) devices have chiefly been deployed in emergency medicine, where they are considered a valid tool. The data situation is less clear in the case of internal questions in abdominal sonography. In our study, we investigate whether HHUS devices from different manufacturers differ in their B-scan quality, and whether any differences are relevant for the significance of an internal ultrasound examination. Method: The study incorporated eight HHUS devices from different manufacturers. Ultrasound videos of seven defined sonographic questions were recorded with all of the devices. The analogue recording of the same findings with a conventional high-end ultrasound (HEUS) device served as an evaluation criterion. Then, the corresponding findings were played side by side and evaluated by fourteen ultrasound experts using a point scale (5 points = very good; 1 point = insufficient). Results: The HHUS devices achieved relatively good results in terms of both the B-scan quality assessment and the ability to answer the clinical question, regardless of the manufacturer. One of the tested HHUS devices even achieved a significantly (*p* < 0.05) higher average points score in both the evaluation of B-scan quality and in the evaluation of clinical significance than the other devices. Regardless of the manufacturer, the HHUS devices performed best when determining the status/inferior vena cava volume and in the representation of ascites/free fluid. Conclusion: In various clinical abdominal sonography questions, HHUS systems can reliably reproduce findings, and are—while bearing their limitations in mind—an acceptable alternative to conventional HEUS systems. Irrespective of this, the present study demonstrated relevant differences in the B-scan quality of HHUS devices from different manufacturers.

## 1. Introduction

Medical ultrasound (US) plays an important role as a ubiquitously available, radiation-free imaging diagnosis method in almost all clinical disciplines [1]. There are many factors that can influence the quality and significance of a US examination. In addition to examiner- and patient-dependent factors, this also includes the technical quality of the ultrasound device used.

Currently, devices of different quality classes can be used for ultrasound diagnostics [2]. High-priced, high-end ultrasound (HEUS) devices are distinguished from mid-range devices and mobile handheld ultrasound (HHUS) devices [3]. Although quality differences between these grades are known [4], there is no sharp dividing line between them.

Even though portable and bedside US devices have existed for more than 40 years [5], the use of HHUS devices in gown pocket format has only become increasingly widespread in recent years [6,7] due to improvements in technology and a reduction in device size [1]. In addition to the hardware of an HHUS device, the software also plays an important role in the optimal image construction and timely conversion of the acoustic ultrasound signals into a two-dimensional B-scan [1]. Market analyses confirm that HHUS has significant further growth potential [8]. Therefore, evidence-based data are all the more important for assessing the technical quality of HHUS compared to conventional HEUS.

There are numerous studies that define suitable areas of application for HHUS. Among other things, its use in emergency medicine or POCUS has been evaluated [9,10,11,12]. Due to the small size of the devices, HHUS also became established in outpatient care [13,14], under isolation conditions [15] and in preclinical emergency medicine [16,17]. Other areas of application evaluated include geriatrics [14], palliative medicine and supply in rural and underserved communities [18,19].

In several studies, HHUS was investigated in a comparison with conventional HEUS with regard to different clinical questions. The conclusions of these studies are highly heterogeneous, and describe both inadequate qualities [13,20] and very good qualities [2,21,22] in frequently occurring internal medicine questions.

There are at least eleven competing manufacturers of HHUS systems on the European medical device market whose clinical outcome quality has not yet been systematically compared. Only one recently published study compared the HHUS devices of different manufacturers [23]. In this study, four different devices were successively assessed by multiple examiners on one subject for three clinical questions with regard to manageability and imaging quality. One of the four devices tested was rated significantly worse than the other three.

Objectively recording B-scan quality is difficult and not standardised. In the present study, B-scans by HHUS devices were evaluated by multiple highly experienced examiners via a direct visual comparison with B-scans from standard US devices.

The aim of the present study is to assess and compare the current HHUS models from eight different manufacturers with regard to their B-scan quality. Finally, in a further step, the question of whether there are differences with respect to clinical significance or pathology presentation is examined.

## 2. Materials and Methods

### 2.1. Study Design

This clinical non-inferiority study was designed prospectively, and serves to compare a variety of HHUS devices from different manufacturers. An HEUS device served as the reference device. The primary endpoint of the study was the evaluation of B-scan quality using a five-point evaluation scale. Secondary endpoints relate to the possibilities of answering clinical questions, also using a five-level assessment scale. The assessment was carried out by DEGUM-certified ultrasound specialists.

The study protocol was reviewed and approved by the relevant ethics committee of the Theodor Fontane Brandenburg Medical School under number E-01-20220502. The patients involved provided their written consent to the use of their anonymised findings for this study, in accordance with the study protocol.

The study incorporated portable, ultra-compact sonography devices, which without a monitor are not significantly larger than a conventional convex transducer. The devices are battery-powered and can be guided with one hand [4]. With these devices, the images are displayed on a separate monitor (a “tablet” computer, mobile phone or iPad). The images produced are transmitted either via a cable or via a WLAN connection. The devices are to be approved for medical B-scan sonography.

### 2.2. Course of the Study

Between July and November 2022, a total of eight HHUS devices from different manufacturers were deployed under clinical conditions. Figure 1 shows the course of the study, including the methodology implemented.

We defined seven typical sonographic questions that can be answered in everyday clinical practice in an internal medicine clinic with the help of routine sonography on patients. An HEUS device (Canon Aplio i900, Canon Medical Systems Corporation, Ōtawara, Japan), which was defined as the “gold standard”, was used for this purpose. Immediately afterwards, a follow-up examination of the findings was carried out by the same examiner using an HHUS device.

The findings for both devices were documented in the form of a short video clip. Following digital anonymisation by the IT Department of the institution with respect to patient and manufacturer information, these corresponding video clips were displayed side by side on a screen (evaluator-blinded).

The comparative evaluation was carried out afterwards by experienced ultrasound examiners. In an online interview, both video clips were played side by side (Figure 2).

First, the B-scan quality of the HHUS device was assessed, using the parallel video clip of the same finding as recorded with the HEUS device as a benchmark. The assessment was made using a five-point Likert scale (very good–good–acceptable–sufficient–insufficient).

In the second step, the clinical significance of the video was assessed with regard to the sonographic question applied to the HHUS device. The above five-point Likert scale was also used for this purpose.

All eight HHUS devices tested were assessed on the basis of the seven previously defined sonographic questions compared to the high-end device with regard to B-scan quality, and then with regard to clinical significance.

The results from the Likert scale were converted into a point system. For this, 5 points were allocated for “very good”, 4 points for “good”, etc., and only 1 point for “insufficient”. In this way, an average score was cumulatively determined for each device and for each sonographic question, the level of which correlated with the B-scan quality or the clinical significance. This score enabled a comparative assessment of B-scan quality and clinical significance, as well as the creation of a ranking of the HHUS devices used.

### 2.3. Devices Used

At the time of the study, eleven different manufacturers of HHUS devices were identified in the German medical device market. All of the manufacturers were contacted in writing and asked to provide a test sample of their latest model. Eight manufacturers provided a positive response and provided appropriate equipment in the form of a loaner (Table 1).

It was not possible to include three devices in the study due to a lack of response from the manufacturer or because no devices were available at the time of the study. Three of the eight devices included worked by means of wireless image transmission, and five were connected to a tablet or a mobile phone through a cable. Two devices were tied to an Android operating system; for the rest, it was possible to install device-specific software on Windows, Android and iOS tablets. All eight devices worked with tissue harmonic imaging (THI) for B-scan optimisation, and all devices also had a duplex module as well as various options for data storage (DICOM: digital imaging and communications in medicine) on their hard drive or in a corresponding cloud. Some of the devices had one or two control buttons, which were variably programmable (e.g., with the “Freeze” function), and other devices could only be operated via their screen. The HHUS devices we used were equipped with either curvilinear or phase arrays, operated at sound frequencies between 2.5 and 5.5 MHz, and used piezoelectric crystals as transducers. Only the Butterfly iQ+ (Butterfly Network Incorp) uses selectively controllable microchip elements, allowing a higher frequency range [24]. Differences in the technical or software equipment of the individual HHUS devices will not be described in greater detail here, as this study focuses on the assessment of the B-scan representation of the individual devices. We refer to previous publications in this regard [1,17].

### 2.4. Examiners

All sonographic examinations were carried out by two highly experienced DEGUM-certified ultrasound examiners (level 2 and 3 of the German Society for Ultrasound in Medicine). Both examiners were specialists in internal medicine and had already made more than 5000 findings in abdominal ultrasound. The investigation was highly standardised. To further reduce bias, we tried to bring both examiners to the same level regarding their previous knowledge. Before the deployment of the various HHUS devices, both examiners studied in detail the device-specific application modes and their options for image optimisation. Both investigators had used HHUS on an infrequent basis prior to the study, and had no specific/detailed previous experience with any of the devices.

### 2.5. The Patients

The findings were collected from patients who had indications for abdominal sonography as part of inpatient treatment at an internal medicine clinic. A total of forty-eight patients participated in the study. The average age of the patients was 71.3 years (95% confidence interval 67–75.4) and they had an average BMI of 25.6 kg/sqm (95% CI 24.7–26.5). Twenty-six of the patients were female and twenty-two were male.

### 2.6. Sonographic Questions

The sonographic questions used are intended to reflect typical questions in clinical internal medicine. Documentation of the findings was based on the specifications of the national and international professional associations [25,26]. The sections, transducer positions and sonographic questions used are listed in Table 2.

The selection encompassed questions concerning which interfaces with large impedance jumps occur (ascites, volume status/IVC), as well as those that require the mapping of low greyscale gradations (e.g., pancreas, liver lesion). It encompassed incisions with a low penetration depth (needle tracking in situ, pancreas) as well as those with a high penetration depth (volume status/IVC, ascites). Needle tracking in vivo occupies a special position. Here, a gelatine model on which a 1.5 cm large round mass is punctured free-hand using a Tru Cut biopsy needle 16 G while being sonographically controlled is used. As an example, we list the representations of needle tracking in situ using eight different HHUS devices (Figure 3).

### 2.7. Evaluators

The assessment was carried out by a total of fourteen doctors highly experienced in clinical ultrasound. All of the evaluators were certified according to the DEGUM criteria (level I: *n* = 1; level 2: *n* = 7; level 3: *n* = 5). All evaluators were specialists in internal medicine, nine of them were additionally specialists in gastroenterology, and one each were specialists in pulmonology, nephrology, emergency medicine and diabetology (multiple responses possible). Five of the evaluators worked on an outpatient basis, and the remainder worked predominantly in the inpatient setting of an internal medicine clinic.

### 2.8. Statistics

The data were collected using Microsoft Excel^®^ version 16.48 (Microsoft Corporation, Redmont, WA, USA). SPSS software version 22.0 (IBM SPSS Statistics ^®^, New York, NY, USA) was used to analyse the data.

The individual HHUS devices were assessed both in terms of their B-scan quality and in terms of clinical significance using a five-point Likert scale. Each evaluator allocated an exact point value for each device and for each sonographic question (5 points for “very good”, 4 points for “good”, 3 points for “acceptable”, 2 points for “sufficient” and 1 point for “insufficient”). From the point values obtained, average values could be formed both for the overall result and for the subgroup analysis, which make the assessments numerically assessable by the evaluators.

The binary and categorical baseline variables are stated as absolute numbers and percentages. Continuous data are stated as mean and standard deviation (SD). Statistical significance was evaluated using the exact Wilcoxon rank-sum test for continuous variables. All significance tests were performed bilaterally. *p*-values < 0.05 were considered statistically significant.

## 3. Results

### 3.1. B-Scan Quality

The results of the evaluated B-scan quality can be found in Table 3, Appendix A. The HHUS devices used scored in terms of their B-scan quality between 2.59 ± 1.06 (Device H) and 3.83 ± 0.77 (Device A). The best-rated device (Device A) showed a significantly better rating than the other seven tested devices (Devices B–H) (Figure 4).

For the best-rated device (Device A), the B-scan quality was rated “acceptable” or better in 96% of the reviews submitted. This was the only device for which an “insufficient” rating was not allocated (Figure 5a). The worst-rated HHUS device scored “good” or “very good” in only 13% of the assessments.

In the B-scan quality assessment as a function of the sonographic questions, the representations of ascites, volume status/IVC and “needle in vitro” achieved better scores than the remaining four questions applied (Figure 4b).

### 3.2. Clinical Significance

The results of the evaluated clinical significance can be found in Table 4 and Appendix A. The HHUS devices used scored between 2.79 ± 1.06 (Device H) and 3.97 ± 0.93 (Device A) in terms of clinical significance. In this test too, the best-rated device (Device A) showed a significantly better point score than the next-best device (here, Device D) (Figure 6a). However, the difference was not as great as in the evaluation of the B-scan quality.

For the best-rated device (Device A), the B-scan quality was rated “acceptable” or better in 96% of the reviews submitted; almost two-thirds of the reviews (64%) were “good” or “very good”. This was the only device for which an “insufficient” rating was not allocated (Figure 6b).

The worst-rated HHUS device scored “good” or “very good” in only 29% of the assessments.

In the assessment of the devices with regard to clinical significance, the representations of ascites, volume status/IVC and “needle in vitro” achieved significantly better scores than the remaining four questions applied (Figure 6b).

### 3.3. Comparison of B-Scan Quality vs. Clinical Significance

As expected, the B-scan quality scores achieved correlated with the scores for clinical significance. Overall, the ability to answer clinical questions was rated somewhat better than the B-scan quality assessment (Appendix A).

### 3.4. Designation of Manufacturers

Device A achieved the best point score both in terms of B-scan quality and in terms of clinical significance. It is the Vscan Air from General Electric Healthcare, Chicago, IL, USA. The second-best device in the evaluation of the B-scan quality was Device B, which is Butterfly iQ+ from Butterfly Network, Inc., Burlington, MA, USA. In terms of clinical significance, Device D ranked second; this was Kosmos from EchoNous, Inc., Redmond, WA, USA.

## 4. Discussion

Progressive technical development with the increasingly smaller size of devices and increasing digitisation has led to sonography devices continually becoming smaller and more manageable, while at the same time being able to demonstrate astonishing imaging quality [1,22,27].

The quality of an abdominal ultrasound examination depends on many different factors. In addition to examiner-dependent and patient-dependent factors, device quality plays a decisive role. Here, the term device quality is only defined vaguely; it can refer to technical imaging accuracy in standardised dummy models or to the subjective “impression” of the examiner during the clinical application of the ultrasound.

In order to objectively capture the imaging quality of B-scans, phantom models that are also commercially available were developed [28,29]. These phantom models were unable to establish themselves as the standard for sonography studies. Phantom models are accepted for functional control in the case of technical defects [30,31,32], and have also been used in technical innovations on sonography devices [33,34,35]. Currently, the image quality of any new US method is routinely evaluated, in terms of contrast, signal-to-noise ratio and resolution, by objectively measuring suitable parameters on images produced by US phantoms [36,37]. Nevertheless, we favoured a subjective approach that is closer to the clinical application of the tested scanners

Most previous examinations for the evaluation of technical innovations, such as tissue harmonic imaging [38,39,40,41,42] or photopic imaging, [43,44] but also comparative examinations of similar devices [45,46], evaluate them on the basis of subjective impression; in these, multiple “experienced” examiners usually carry out evaluations in parallel. This study also captured the subjective impressions of experienced examiners. In order to capture a broad database with data that are as objective as possible against this background, fourteen experienced examiners participated in the evaluation. In order to capture the subjective impression numerically, we used [41,45,47] a five-level Likert scale, as in previous studies. A point system was used to quantify and compare the quality.

### 4.1. Comparison of HHUS Devices and HEUS Devices

The increasing clinical use of HHUS devices has been observed since the early 2000s and evaluated accordingly [2,45,48]. Application studies exist for HHUS systems in various disciplines [16,49,50], all of which used high-end devices as a comparison. In this context, in 2019, a meta-analysis appeared [51] in which a total of sixteen HHUS vs. HEUS comparative studies were included. Of these, only two were related to global abdominal sonography [13,21] and three to defined questions of internal abdominal sonography [52,53,54].

### 4.2. Assessment of HHUS Based on Clinical Questions

In most studies, the quality of the presentation of findings does not extend to the image quality of HEUS. This is not surprising due to the compactness of HHUS. Therefore, the question arises as to whether and to what extent HHUS is able to positively influence clinical and preclinical decision paths. This could be demonstrated in several and, in some cases extensive, examinations, e.g., in palliative medicine [18], in geriatrics [19], in traumatology [50,55] and in various questions concerning emergency medicine [56,57,58]. Here, an extensive examination of HHUS in emergency medicine, which not only proves a safe and faster diagnosis through the use of HHUS, but also assumes a reduction in in-hospital mortality, should be specifically highlighted [12].

### 4.3. Comparison of Similar Devices

Only a few studies carry out a head-to-head comparison of comparable US systems from different manufacturers. Various endosonography systems are described in the pancreatic assessment [46], and in a further publication, the ultrasound quality of various US devices using contrast agent in inflammatory bowel diseases [59] is estimated. Beyond the B-scan assessment, there are comparisons with respect to the ergonomics of ultrasonic systems [60] and several comparisons of the shear wave elastography of different devices [34,61,62]. Extensive comparisons of the B-scan quality in abdominal sonography on high-end devices and mid-range sonography devices have been carried out by our own working group [3,63].

A first comparison of HHUS devices from different manufacturers in terms of technical equipment, manageability and image quality appeared only recently [23]. In this, a total of four different HHUS devices were compared with each other for three different clinical questions. The questions were related to aspects of emergency medicine. All of the devices used in that study are also included in the study presented here. If one looks at the assessed B-scan quality of HHUS devices in this study, the results only partially correlate with the results we determined. The results are difficult to compare with different study designs, and especially with different sonographic questions. In the study cited, for example, only emergency medical aspects (FAST, duplex US and echocardiography) are considered, while our investigation focused on B-scan quality and various internal sonographic questions.

The present study combines all three of the comparison principles described above. We assessed the B-scan quality of the HHUS devices in a direct visual comparison with an HEUS device. A comparison of the different HHUS devices was made possible due to the fact that a total of eight different HHUS devices were evaluated on the same sonographic questions. Finally, the investigation also assessed the clinical significance and whether the clinical question could be answered with sufficient certainty. This not only makes it possible to rank the HHUS devices used in terms of B-scan quality, but also to assess the clinical relevance of the differences determined.

### 4.4. Technical Configuration of the Devices

Devices can also [23,51,60] differ significantly in terms of ergonomics and manageability. While the devices used by us differed insignificantly in terms of handling or weight, we did notice differences in battery life and the stability of the connection to the monitor. In principle, all eight HHUS devices showed a high and intuitive level of user-friendliness, as well as agreeable monitor quality. However, all of these criteria were not part of the comparison in question, and were therefore not systematically recorded. In this regard, we refer to two extensive analyses by Malik [17] and von Dietrich et al. [1]. While image optimisation in conventional stand-alone devices can almost always be achieved at a low threshold with the help of rotary or slider controls, in the case of HHUS devices, this can only be achieved using digital software applications due to their ultra-compact design. As this makes readjusting the image processing algorithm more challenging, it is necessary to primarily rely on the settings present in the provided presets. It should be noted that the quality of the resulting image does not only depend on the device itself, but also on the software package used to create the images. For this reason, the HHUS devices used in the study were each equipped with the latest available software packages. Image processing algorithms influence B-scan quality and clinical interpretation in handheld ultrasound. Both beamforming techniques [64,65] as well as adjusting tissue attenuation [66] are widely recognised as crucial aspects in producing accurate B-scans. It is reasonable to assume that the HHUS devices utilised these techniques and considered them in the various presets. Manual readjustment may not have been possible or entailed navigating through several sub-levels. For instance, the positioning of the focus was only visible on the monitor for one (Philips Lumify, Philips, Amsterdam, The Netherlands) of the eight HHUS devices utilised, and was amendable through manual adjustment.

Each of the devices used has specific technical possibilities for image optimisation, which can have a considerable influence on image quality [67]. Therefore, the examinations were carried out by two examiners, both of whom were certified according to DEGUM criteria and could call on intensive, long-term experience with different ultrasound systems. Therefore, this minimised the disadvantages of individual devices due to possible operating errors.

### 4.5. Cohort and the Clinical Questions Used

In addition to age, gender and constitution, there are various other subject-dependent influences on the imaging quality of an ultrasound examination. The cohort we used was recruited from the real patient population of an internal medicine clinic. The cohort covered both sexes equally and, and had an average age of 71.3 years and a BMI of 25.6 kg/sqm, which likely provided a quite realistic representation of typical patients in an internal medicine clinic.

A highly objective comparison of the eight different HHUS devices under clinical conditions would have been possible if the same clinical findings were examined successively with all eight HHUS devices that were to be tested. In this case, different patient-related or constitutional sound conditions, such as obese patients or patients with challenging anatomy, would not be significant. For logistical reasons, this procedure was not feasible, as we never had more than two different HHUS devices at our disposal at any one time. In order to ensure that differences in the sonicity of individual patients were not very important in the evaluation, we did not estimate the absolute image quality of the HHUS devices, but used the video from an HEUS device, recorded in parallel, as an evaluation criterion. Therefore, the assessment of the HHUS devices is not absolute, but made in relation to the parallel recorded findings of the HEUS device. If, for example, difficult sound conditions existed in the case of obesity or long start-up length, these would have an influence on the B-scan quality of both devices, and would therefore not disadvantage the HHUS device due to inferior sound conditions. Nevertheless, future studies should address these issues and, if possible, use multiple devices on the same patient.

By selecting different sonographic questions on different subjects, we attempted to map a variety that simulated the clinical reality of an internal medicine clinic. In contrast to the previous comparative examination of high-end sonography devices by our working group [63], we were able to include frequent clinical questions and predominantly pathological findings in the current examination. By selecting the sectional planes, we attempted to take the HHUS devices to their limits, for example, through high penetration depth or through increased requirements in the separation of small impedance jumps. Of the seven previously defined sonographic questions, only two related to physiological findings. These two sections were intended to describe the volume status on the basis of the inferior vena cava, whereby this was to be represented with the liver as a sound window on the transducer. As a second physiological section, we chose the pancreas, which is not always very easy to represent due to overlap with the stomach and, in some cases, the large intestine. The pathological findings used (Table 2) correspond to very common questions in internal medicine. Needle tracking—both in situ and in the gelatine model—is a special case. With this, we wanted to include interventional US, and are aware that this question is more relevant for specialist sonography departments in hospitals than, for example, in examinations in an outpatient setting.

### 4.6. B-Scan Quality and Correlation to Clinical Questions

Of the HHUS devices tested, Device A (Vscan Air, GE) achieved significantly better scores than the other devices, both in terms of B-scan quality and clinical significance. With the “Vscan”, GE draws on its many years of experience in the production of HHUS devices. The first publications using Vscan HHUS devices were published more than ten years ago [68]. In the most extensive meta-analysis to date, which examined sixteen publications on the quality of [51] HHUS devices, the GE-Vscan Air device was used fifteen times, from which a high market presence can be concluded. In terms of price, this device is in the mid-range of all of the tested HHUS devices [17].

If one looks at the results of all of the HHUS devices used, the “insufficient” rating was allocated very rarely (Figure 5). As a result, an astonishingly good quality of HHUS devices can be confirmed, which correlates with the results of several previous investigations [22,48,53,69].

The subgroup analysis as a function of the US sections used showed a significantly better evaluation of the HHUS devices in the detection of ascites, in the representation of the inferior vena cava and in needle tracking in the gelatine model. This applied to all of the HHUS devices. It is not surprising that good images are easily generated, both in the case of free liquids and in the assessment of the VCI images, since high impedance jumps, which can also be easily detected by a less demanding US technology, are present here. Needle tracking in the gelatine model is the only in vitro image to occupy a special position in the US sections used, as artefacts from connective tissue or air do not occur here. This should make the sonographic representation considerably easier. The situation is quite different in the representation of the gallbladder and, above all, of the pancreas, since both connective tissue and air overlays have to be overcome, and small impedance jumps have to be mapped. The representation of a puncture needle in situ was also somewhat more difficult with the HHUS devices than with the other sonographic questions mentioned above.

These results correlate well with the experiences of several previous investigations, which confirm the good imaging quality of HHUS devices, especially in the case of findings with high impedance jumps, such as in the representation of ascites, pleural effusions, urinary bladder volume and urinary congestion, but also in the case of aortic aneurysms and in echocardiography [52,70,71,72]. In our study, the B-scan quality for the assessment of hepatic lesions was evaluated to be in the acceptable range. This illustrates the potential of HHUS as a possible screening tool, especially in developing countries [73].

### 4.7. Limitations

The present study has limitations. The most important limitation is that the different HHUS devices were not compared with each other on the same subjects. Therefore, the devices had to prove themselves on the basis of different findings from different subjects. This was at least partially compensated for by the fact that multiple different clinical situations (Table 2) were used in the evaluation of the individual devices. A possible aim for future comparative studies of HHUS devices could be to apply all devices to be tested to the same subjects at the same time.

It is known that imaging quality depends considerably on the device settings [67]. While image optimisation in conventional stand-alone devices can almost always be achieved at a low threshold with the help of rotary or slider controls, in the case of HHUS devices, this can only be achieved using digital software applications due to their ultra-compact design. It should be noted that the quality of the resulting image depends on the device itself, as well as the software package used to create the images. For this reason, the HHUS devices used in the study were each equipped with the latest available software packages. Furthermore, similar presets were used in this study with the involvement of the manufacturers. All of the HHUS devices had a preset for the abdomen that was used as standard. Although image optimisation on the HEUS, such as adjusting the frame rate and dynamic range, can easily be carried out through the manual controls on its panel, readjusting the HHUS device was much more complex, as it necessitated navigating through software submenus. This was the case for all of the HHUS devices we sampled. Although similar presets were used in the study, a limitation could be the different vendor-specific software of the devices. The two very experienced examiners in the present study tried to achieve sufficient image optimisation on all of the HHUS devices used. Possible biases of the ultrasound examiners who created the images cannot be completely ruled out. Furthermore, a possible influence of the transducer shape on the image quality during image generation by the two examiners cannot be completely ruled out. The ease of use and portability of the HHUS devices were not evaluated within the study. Future studies should investigate these two aspects. In the spring of 2022, we were able to identify eleven HHUS systems from different manufacturers that were available in the European market. There may be other manufacturers that we overlooked at this time in our own market research. In addition, the HHUS systems are constantly subject to new technical developments, which can have an impact on the validity of our study.

The test subjects were recruited from the inpatient population of an internal medicine clinic. As a result, the study was based on internal medical questions, and the results described here cannot be easily transferred to outpatients or to patients in the field of emergency medicine.

Another limitation is that the HHUS systems were compared with only one HEUS device, and bias cannot be excluded. Future studies should include multiple HEUS devices. Finally, there was no determination of interobserver and intraobserver reliability, because of the high number of 14 raters involved.

### 4.8. Outlook

Artificial intelligence (AI) has been applied in the field of ultrasound imaging for some time now. Examples of this are modules for assessing liver masses [74], thyroid nodules [75], lymph nodes in breast cancer patients [76] and a meta-analysis with various AI systems for assessing liver fibrosis/cirrhosis [77]. In a first publication, a combination of HHUS devices and AI in particular was used in breast tumour patients [78], and a pilot study on the use of HHUS devices in abdominal sonography was initiated by our working group [79]. In a recent literature search, we were not able to find any publications on AI-based systems for the comparative assessment of B-scan quality in sonography devices; this should be further investigated in future studies. It is possible that artificial intelligence will have an even greater positive impact on B-scan quality and clinical diagnosis via HHUS in the future [80,81]. HHUS has the potential to integrate with other healthcare technologies like telemedicine or electronic medical records and education to enhance patient care. In particular, the telemedicine approach to diagnosis and training should be given greater consideration in the future [82,83].

It would be desirable to establish an independent, superior institution that can provide objective assessments of B-scan quality across the many different HHUS systems. Possibly, this institution could be organised by the relevant professional associations. There are numerous HHUS position papers from various professional societies [4,9,84]. Clear and recognisable indications for the use of HHUS in clinical guidelines are yet to be established. The present study and future research on B-scan quality assessment could contribute to a more reliable definition of the significance of HHUS in clinical practice.

## 5. Summary

The results of this study show that HHUS devices currently available on the market can reliably reproduce findings in various sonographic questions concerning abdominal sonography and—while bearing their limitations in mind—represent an acceptable alternative to conventional HEUS devices. Irrespective of this, the present study was able to demonstrate differences between the tested HHUS devices in terms of B-scan quality and the ability to answer clinical questions.

Future studies should examine further aspects of HHUS devices, e.g., with regard to duplex sonography and echocardiography, but also with regard to a more precise definition of limitations in everyday clinical practice. Given the large number of different HHUS devices available, superordinate quality control of the sonographic imaging quality of HHUS devices would be desirable.

## Figures and Tables

**Figure 1 diagnostics-13-03622-f001:**
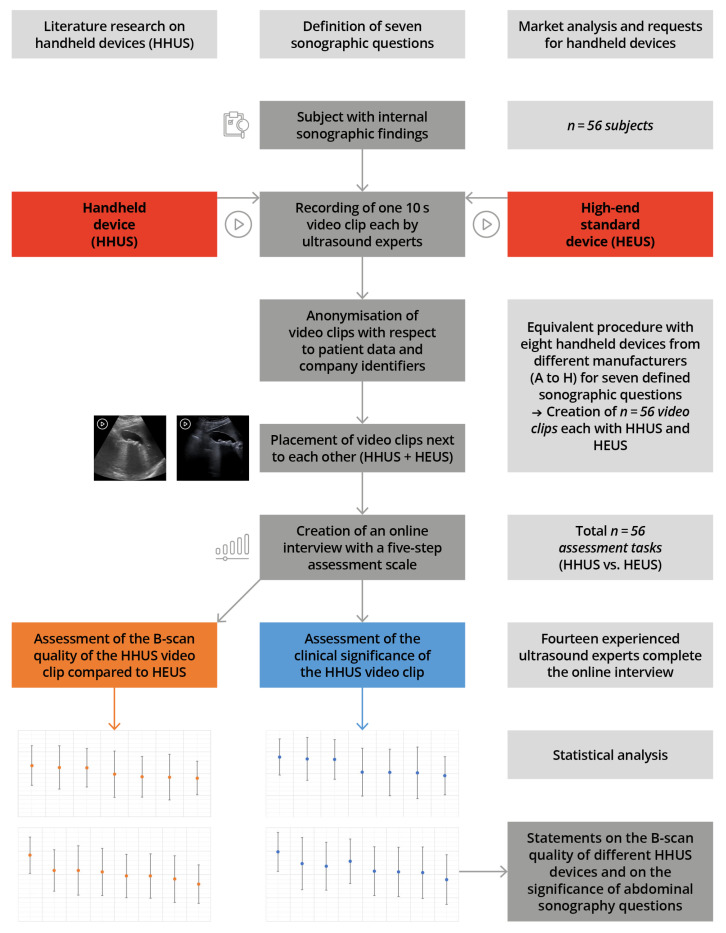
Presentation of the course of the study, including the materials and methodology.

**Figure 2 diagnostics-13-03622-f002:**
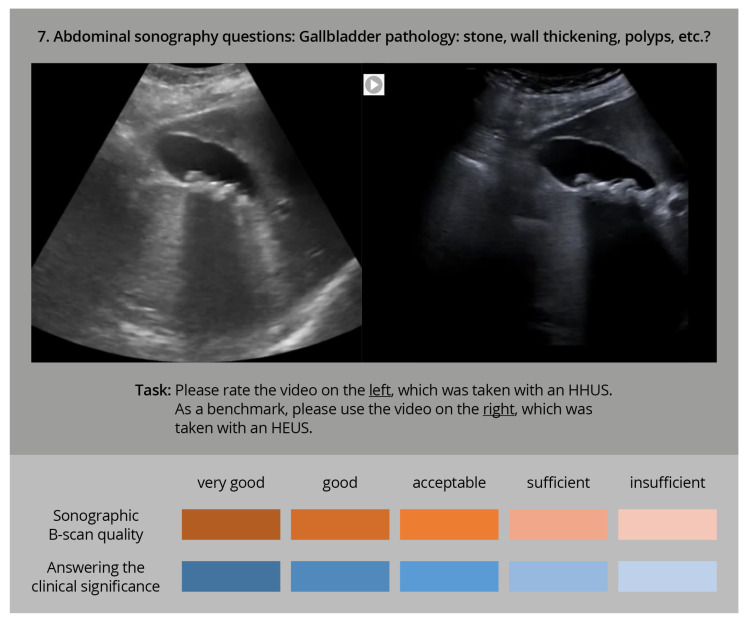
Sample question of the online interview with two sonography videos played in parallel (HHUS and HEUS), to be evaluated by the participating ultrasound experts using two rating scales with regard to sonographic image quality and clinical significance.

**Figure 3 diagnostics-13-03622-f003:**
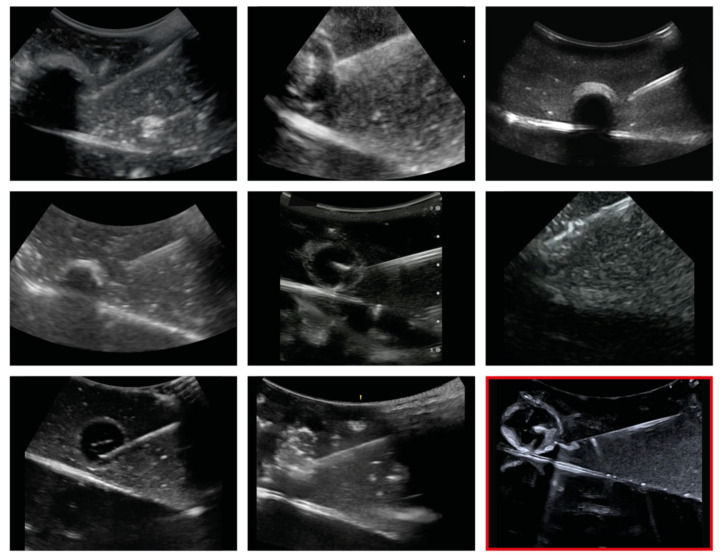
Exemplary representation of the sonographic question “Needle tracking in situ” by eight different HHUS devices (in random order), with the HEUS device (Canon Aplio 900i) used as reference ((**bottom right**)—marked in red).

**Figure 4 diagnostics-13-03622-f004:**
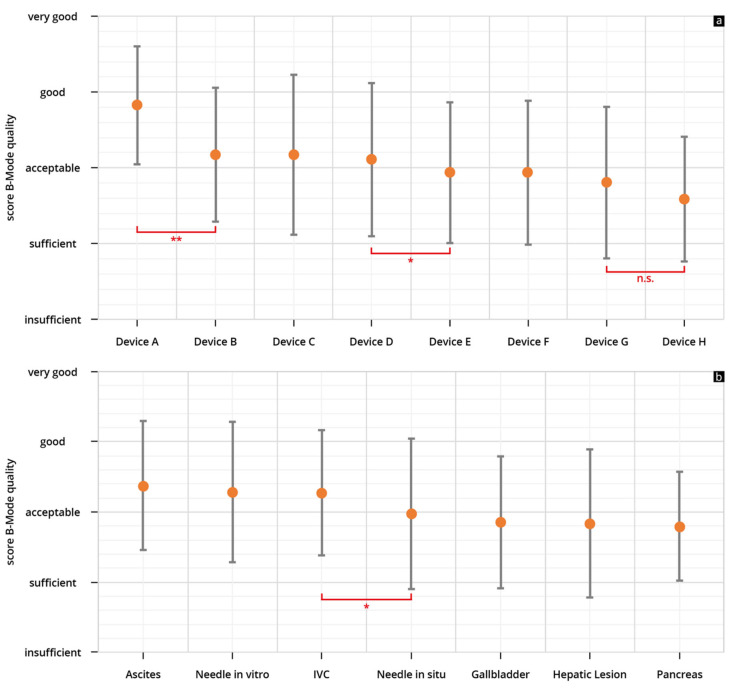
Mean point scores and SD achieved in assessment of the B-scan quality of individual HHUS devices (**a**) and as a function of the sonographic sections (**b**) (* *p* < 0.05; ** *p* < 0.01; n.s. not significant). The order in which the individual HHUS devices are displayed was determined by the score achieved. The device with the highest point score in the evaluation of B-scan quality is referred to as Device A, the following devices are accordingly designated Device B, C, etc., and the device with the worst rating, Device H. A complete list of the *p*-values can be found in the Appendix A (IVC: inferior vena cava).

**Figure 5 diagnostics-13-03622-f005:**
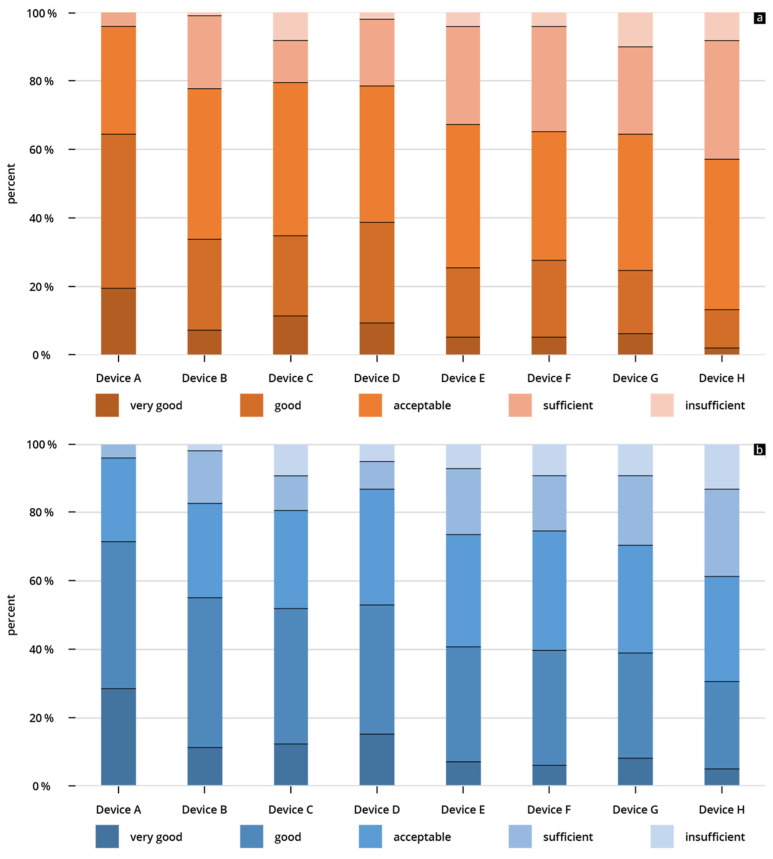
Percentages of given ratings (cumulative) for (**a**) B-scan quality of the individual HHUS devices and (**b**) clinical significance of the individual HHUS devices. A complete list of the *p*-values can be found in Appendix A.

**Figure 6 diagnostics-13-03622-f006:**
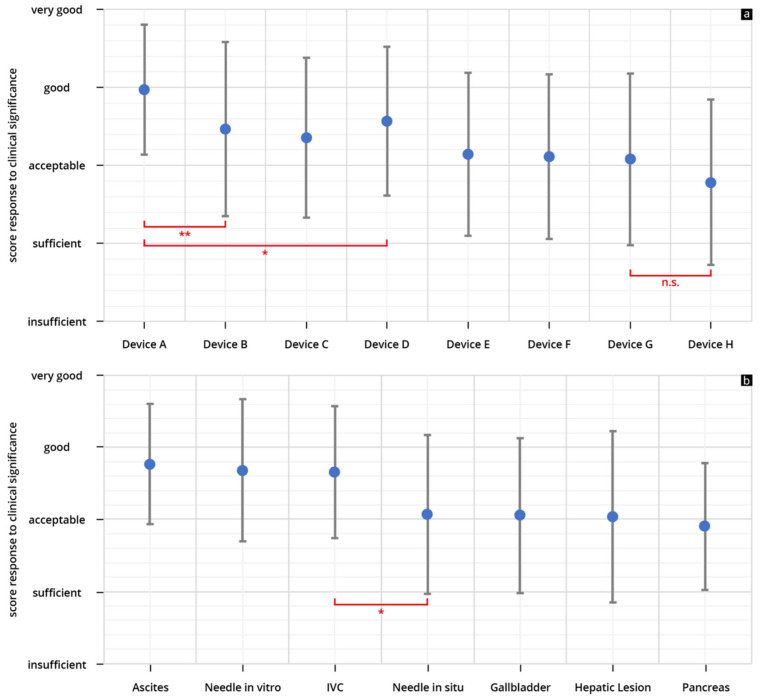
Mean point scores and SD achieved in the assessment of the ability of the individual HHUS devices to answer the clinical question (**a**) and as a function of the sonographic sections (**b**) (* *p* < 0.05; ** *p* < 0.01; n.s. not significant). The order in which the individual devices are displayed is as shown in Figure 4, depending on the score achieved in the evaluation of the B-scan quality. A complete list of the *p*-values can be found in Appendix A (IVC: inferior vena cava).

**Table 1 diagnostics-13-03622-t001:** Overview of the HHUS devices requested for the study (in alphabetical order); the devices marked with * could not be included in the study due to a lack of response from the manufacturer or because no devices were available at the time of the study.

Device Name	Manufacturer	City, Country	Image Transmission	Transducer
Alpinion minisono	Alpinion Medical Systems	Seoul, Republic of Korea	Wired	Convex or linear
Butterfly iQ+	Butterfly Network, Inc.	Burlington (MA), USA	Wired	“1.75 D-Array”
Clarius C3 HD3	Clarius Mobile Health Corp.	Burnaby (BC), Canada	Cordless	Multiple attachable probes
iSiniQ 30A *	Prunus Medical Shenzhen	Shenzhen, China	Cordless	
Kosmos	EchoNous Inc.	Redmond (WA) USA	Wired	All-in-one transducer
mSonics MU 1 *	Youtu Technology (Zhuhai) Co., Ltd.	Guangdong, China	Unknown	
Philips Lumify	Koninklijke Philips N.V.	Amsterdam, The Netherlands	Wired	Broadband convex and two other variants
SonoSite iViz	FUJIFILM Corporation	Tokio, Japan	Wired	All-in-one transducer
Sonostar * Uprobe-C4PL	Universal Diagnostic Solutions	Vista (CA), USA	Cordless	Convex and linear in one
Vscan Air	General Electric	Boston (MA) USA	Cordless	Convex and linear in one
Youkey Q7	Wuhan Youkey Bio-Medical Electronics Co., Ltd.	Wuhan, China	Cordless	Replaceable transducers

**Table 2 diagnostics-13-03622-t002:** Characterisation of the seven sonographic questions used in relation to the organ, section, sonographic field and penetration depth.

No.	Organ	Sonographic Question	Sonographic Section	Near vs. Far Field	Penetration Depth
1.	Ascites	Quantity, echogenicity	Middle abdomen right, Morison’s pouch	Middle field/far field	up to 12 cm
2.	Needle in vitro	Ascites or pleura	Middle abdomen right or intercostal	Near field	up to 6 cm
3.	Vena cava inferior	Diameter vena cava, volume status	Median sagittal section	Far field	up to 12 cm
4.	Needle in situ	Targeted hand puncture	Gelatine model	Near field	up to 4 cm
5.	Gallblad-der	Stone, sludge, wall thickening	Diagonal section right upper abdomen	Near field	up to 6 cm
6.	Liver lesion	Solid mass	Subcostal cut, intercostal cut	Middle field	up to 8 cm
7.	Pancreas	Organ boundaries, Ductus pancreaticus	Epigastric cross-section	Near field	up to 8 cm

**Table 3 diagnostics-13-03622-t003:** Results of the evaluation of B-scan quality with respect to the sonographic question (IVC: inferior vena cava).

Device	Gall-Bladder	Ascites	IVC	Needle In Vitro	Pancreas	Hepatic Lesion	Needle In Situ	Total Score
	MW ± SD	MW ± SD	MW ± SD	MW ± SD	MW ± SD	MW ± SD	MW ± SD	MW ± SD
A	3.71 ± 0.61	4.00 ± 0.78	4.07 ± 0.73	4.43 ± 0.65	3.21 ± 0.58	3.71 ± 0.83	3.64 ± 0.74	3.83 ± 0.77
B	3.43 ± 0.85	3.50 ± 1.02	3.21 ± 0.80	3.07 ± 1.00	3.14 ± 0.66	2.43 ± 0.51	3.43 ± 0.94	3.17 ± 0.89
C	1.93 ± 0.83	3.86 ± 0.77	2.93 ± 1.00	2.86 ± 0.95	3.43 ± 0.76	3.50 ± 0.76	3.71 ± 1.07	3.17 ± 1.06
D	3.43 ± 0.65	3.21 ± 0.70	3.36 ± 0.74	2.86 ± 0.77	2.36 ± 0.50	3.79 ± 0.70	2.79 ± 1.19	3.11 ± 1.01
E	3.36 ± 0.74	2.57 ± 1.02	3.43 ± 0.65	3.79 ± 0.70	2.93 ± 0.73	2.36 ± 0.50	2.14 ± 0.86	2.94 ± 0.93
F	2.14 ± 0.66	3.79 ± 0.70	3.57 ± 0.76	2.93 ± 0.83	2.64 ± 0.63	3.36 ± 0.84	2.14 ± 0.77	2.94 ± 0.95
G	2.93 ± 0.62	3.21 ± 0.80	2.43 ± 0.85	3.64 ± 0.93	2.50 ± 0.65	1.50 ± 0.65	3.43 ± 0.76	2.81 ± 1.10
H	2.71 ± 0.73	2.86 ± 0.66	3.14 ± 0.77	2.71 ± 0.91	2.14 ± 0.77	2.07 ± 0.73	2.50 ± 0.76	2.59 ± 1.06

**Table 4 diagnostics-13-03622-t004:** Results of the evaluation of clinical significance with respect to the sonographic question (IVC: inferior vena cava).

Device	Gallbladder	Ascites	IVC	Needle In Vitro	Pancreas	Hepatic Lesion	Needle In Situ	Total Score
	MW ± SD	MW ± SD	MW ± SD	MW ± SD	MW ± SD	MW ± SD	MW ± SD	MW ± SD
A	3.86 ± 0.66	4.29 ± 0.61	4.29 ± 0.83	4.50 ± 0.65	3.29 ± 0.61	4.14 ± 0.86	3.43 ± 0.85	3.97 ± 0.93
B	3.79 ± 0.85	3.79 ± 1.02	3.71 ± 0.80	3.79 ± 1.00	2.93 ± 0.66	2.57 ± 0.51	3.71 ± 0.94	3.47 ± 1.11
C	1.86 ± 0.95	4.00 ± 0.68	3.21 ± 0.89	3.00 ± 1.04	3.64 ± 0.74	3.86 ± 0.66	3.93 ± 1.14	3.36 ± 1.02
D	3.64 ± 0.74	3.79 ± 0.70	4.21 ± 0.80	3.29 ± 1.07	2.86 ± 0.66	4.07 ± 0.62	3.14 ± 1.03	3.57 ± 0.93
E	3.36 ± 0.93	2.93 ± 1.14	3.43 ± 0.65	4.21 ± 1.07	2.86 ± 1.03	2.43 ± 0.51	2.57 ± 1.02	3.14 ± 1.05
F	2.14 ± 0.86	4.00 ± 0.51	3.64 ± 0.74	3.36 ± 0.74	2.79 ± 0.80	3.71 ± 0.83	2.14 ± 1.03	2.14 ± 1.03
G	3.00 ± 0.86	3.79 ± 0.55	2.86 ± 0.74	4.14 ± 0.74	2.64 ± 0.80	3.64 ± 1.03	3.64 ± 1.03	3.08 ± 1.10
H	2.79 ± 1.12	3.57 ± 0.51	3.64 ± 0.84	3.14 ± 0.95	2.29 ± 0.99	2.79 ± 1.06	2.07 ± 0.73	2.79 ± 1.06

## Data Availability

The data presented in this study are available on request from the corresponding author. The data are not publicly available due to because of institutional and national data policy restrictions imposed by the ethics committee, since the data contain information that could potentially identify study participants. Data are available upon request (contact via weimer@uni-mainz.de) for researchers who meet the criteria for access to confidential data (please provide the manuscript title with your enquiry).

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
