# Peer review of "Prospective Comparison of Handheld Ultrasound Devices from Different Manufacturers with Respect to B-Scan Quality and Clinical Significance for Various Abdominal Sonography Questions"

_diagnostics, 2023, doi:10.3390/diagnostics13243622_

Round 1

Reviewer 1 Report

Comments and Suggestions for Authors

This study compared several HHUS device in term of the quality of the resut justified by the expert. What should bear in mind is that te quality of the result image is not merely depends on the device itself but also ditermined by the software package used to build the image itself. Authors should mention this crucial factor in order to justify that the result is not different because of different image reconstructoon method used. Authr should also describe the difference factor between HHUS and HHES used in this study

Author Response

Thank you very much for your evaluation of our work.

Please find attached the point-by-point response to the reviewer’s comments.

Reviewer 2 Report

Comments and Suggestions for Authors

 write few analytical questions towards the topic bit difficult questions

1.How does the performance of the two manufacturers' handheld ultrasound devices compare in different patient populations, such as obese patients or patients with challenging anatomy?

2.What is the impact of different transducer types and frequencies on B-Scan quality and clinical significance?

3.How does the interobserver and intraobserver reliability of the two manufacturers' handheld ultrasound devices compare?

4.What is the cost-effectiveness of the two manufacturers' handheld ultrasound devices, taking into account both the initial purchase price and the cost of ongoing maintenance and support?

5.How do the two manufacturers' handheld ultrasound devices compare in terms of their ease of use and portability?

6.Can machine learning be used to improve the B-Scan quality and clinical significance of handheld ultrasound devices?

7.How can handheld ultrasound devices be integrated with other healthcare technologies, such as telemedicine and electronic health records, to improve patient care?

8.What are the ethical implications of using handheld ultrasound devices in point-of-care settings, such as rural and underserved communities?

Comments on the Quality of English Language

moderate english corrections are needed

Author Response

(The authors gave the same response as above.)

Reviewer 3 Report

Comments and Suggestions for Authors

In this paper, the imaging quality of 8 Handheld ultrasound (HHUS) scanners by different manufacturers is compared to that of a reference high-end US scanner. The corresponding videos obtained by 2 experienced sonographers on different subjects were evaluated by 14 US experts.

As a result, the HHUS scanners turned out to provide, on average, good-quality images with significant differences among the different scanners (one was superior).

The paper is very well written (although the Discussion could be shortened a little bit) and addresses a topic (the practical use of HHUS scanners) of increasing interest. The authors admit important limitations of their work (overall: the different scanners were not tested on the same subjects; the image quality could be biased by scanner settings “arbitrarily” chosen by the operators), but do not mention a possible bias by the sonographers who produced the images. Did any of them have previous experience with any of the tested HHUS scanners? This must be clearly stated in the paper.

Furthermore, on p.13, the possible use of phantoms for image quality evaluation is dismissed, based on old references such as 28, 35, and 36. The image quality of any new US method is today routinely evaluated in terms of contrast (see: A. Rodriguez-Molares et al., "The Generalized Contrast-to-Noise Ratio: A Formal Definition for Lesion Detectability," in IEEE Transactions on Ultrasonics, Ferroelectrics, and Frequency Control, vol. 67, no. 4, pp. 745-759, April 2020, doi: 10.1109/TUFFC.2019.2956855.), signal-to-noise ratio, and resolution (which is usually measured through the full-width half-maximum of a point-scatterer echo, see: Thomas L. Szabo, Diagnostic Ultrasound Imaging, 2004), by objectively measuring suitable parameters on images produced by US phantoms. Hence, rather than describing as inadequate the use of phantoms, the authors should simply claim that they preferred a subjective approach, closer to the clinical application of the tested scanners.

As a minor note: on p.14 line 333, what is “the contrast agent ultrasound quality”? Please rephrase

Author Response

(The authors gave the same response as above.)

Round 2

Reviewer 2 Report

Comments and Suggestions for Authors

1. What are the effects of tissue attenuation and beamforming techniques on B-scan quality in handheld ultrasound devices?

2.How do image processing algorithms influence B-scan quality and clinical interpretation in handheld ultrasound?

3. What are the standardization challenges in assessing B-scan quality across different handheld ultrasound devices?

4. How can we effectively integrate B-scan quality data into clinical decision-making for handheld ultrasound applications?

5. Analytical perceptives are needed for more clariity.

Comments on the Quality of English Language

Can be improved

Author Response

Dear Rewiever,

thank you very much for the new evaluation of our manuscript.  We are submitting a second revised version of our manuscript and a  point-by-point response to the reviewer’s comments, which we have thoroughly updated according to your suggestions. 

Best regards
